# PeerJ

# DrugOn: a fully integrated pharmacophore modeling and structure optimization toolkit

Dimitrios Vlachakis[1,2,4], Paraskevas Fakourelis[1,2,4],
Vasileios Megalooikonomou[2], Christos Makris[2] and Sophia Kossida[1,3]

[1] Bioinformatics & Medical Informatics Team, Biomedical Research Foundation, Academy of Athens, Athens, Greece
[2] Computer Engineering and Informatics Department, University of Patras, Patras, Greece
[3] IMGT, Laboratoire d'ImmunoGénétique Moléculaire, Institut de Génétique Humaine, Montpellier, France
[4] These authors contributed equally to this work.

## ABSTRACT

During the past few years, pharmacophore modeling has become one of the key components in computer-aided drug design and in modern drug discovery. DrugOn is a fully interactive pipeline designed to exploit the advantages of modern programming and overcome the command line barrier with two friendly environments for the user (either novice or experienced in the field of Computer Aided Drug Design) to perform pharmacophore modeling through an efficient combination of the PharmA-COphore, Gromacs, Ligbuilder and PDB2PQR suites. Our platform features a novel workflow that guides the user through each logical step of the iterative 3D structural optimization setup and drug design process. For the pharmacophore modeling we are focusing on either the characteristics of the receptor or the full molecular system, including a set of selected ligands. DrugOn can be freely downloaded from our dedicated server system at www.bioacademy.gr/bioinformatics/drugon/.

Corresponding authors
Dimitrios Vlachakis,
dvlachakis@bioacademy.gr
Sophia Kossida,
skossida@bioacademy.gr

## INTRODUCTION

Fully automated methods of pharmacophore model design can help facilitate the process of modern computer based drug discovery (*Chen et al., 2013*; *Wallach & Lilien, 2009*). Computers gain credibility in the field of computational biology and drug design, as new and more efficient algorithms and pipelines are established (*Donsky & Wolfson, 2011*; *Loukatou et al., 2014*; *Ortuso, Langer & Alcaro, 2006*).

The idea of pharmacophore was first defined by Paul Ehrlich as 'a molecular framework that carries (phoros) the essential features responsible for a drug's (pharmacon) biological activity' back in 1909 (*Ehrlich, 1909*; *Lin, 2000*). According to the recent definition by IUPAC, a pharmacophore model is 'an ensemble of steric and electronic features that is necessary to ensure the optimal supramolecular interactions with a specific biological target and to trigger or block its biological response' (*Wermuth, 1998*).

**How to cite this article** Vlachakis et al. (2015), DrugOn: a fully integrated pharmacophore modeling and structure optimization toolkit.
PeerJ 3:e725; DOI 10.7717/peerj.725

With computer-aided drug design being an integral part of the drug discovery and lead optimization process, pharmacophore models have become a key component in understanding the receptor–ligand interactions. Specifically, pharmacophore models have contributed in evolving the drug design process by shifting the focus from 2-dimensional atoms connectivity to 3-dimensional chemical features arrangement (*Faulon et al., 2008*; *Guner, 2002*; *Balatsos et al., 2012*; *Dalkas et al., 2013*) where the features might be hydrophobic or hydrophilic regions, specific atoms, centers of aromatic or not aromatic rings, positive or negative charges and hydrogen bond donors or acceptors (*Pires, Ascher & Blundell, 2014*; *Zhang et al., 2005*). The 3D pharmacophore modeling methods take into consideration the 3-dimensional structures and binding of receptors and inhibitors, in order to identify areas that are favorable or unfavorable to a specific receptor-inhibitor interaction (*Vlachakis, Tsaniras & Kossida, 2012*; *Vlachakis & Kossida, 2013*). Pharmacophore models contribute to drug discovery by providing a number of benefits, such as the fact that they represent chemical function, valid for the existing bounds as well as for unknown agents. In addition, they are computationally efficient due to their simplicity, which makes them suitable for large scale high throughput virtual screening (*Floris et al., 2011*; *Frommel et al., 2003*; *Vlachakis, Argiro & Kossida, 2013*; *Vlachakis, Karozou & Kossida, 2013*). Finally, they are comprehensive and editable, so the information can be easily traced back by adding or omitting chemical feature constraints. A pharmacophore model can be expressed in two ways: firstly in a ligand-based approach and secondly in a structure-based approach (*Yang, 2010*). A major goal in drug design is to increase potency by optimizing interactions such as the binding of a ligand to its pharmacological target; this requires complementarity of both bonding partners in terms of shape and electrostatics (*Korb et al., 2010*). Pharmacophore models have been already used in a variety of projects in order to exploit their benefits in high throughput virtual screening (*Fei et al., 2013*; *Niu et al., 2013*; *Suresh & Vasanthi, 2010*; *Vlachakis, Kontopoulos & Kossida, 2013*; *Vlachakis et al., 2014*). Pharamcophore models have been successfully used for the identification of human chymase inhibitors (*Arooj et al., 2013*) and for the efficient of overlay of drug-like organic molecules (*Wolber, Dornhofer & Langer, 2006*). The benefits of pharmacophore modeling at computer-aided drug design resulted in the development of a variety of automated tools and applications during the past 20 years (*Vlachakis, Koumandou & Kossida, 2013*; *Vlachakis & Kossida, 2013*; *Vlachakis et al., 2013b*). However the pharmacophore modeling approaches have not reached yet their full potential, as they are limited by a number of obstacles, which are dictated by the ongoing demand for reducing todays very high cost of drug design and drug discovery (*Yang, 2010*; *Vlachakis, Tsiliki & Kossida, 2013*; *Vlachakis et al., 2013c*).

Herein, we introduce DrugOn, a free, open source, unix-based software package for pharmacophore modeling. DrugOn is an interactive platform combining the algorithms of PDB2PQR v.1.8 (*Dolinsky et al., 2007*; *Dolinsky et al., 2004*), Ligbuilder v.1.2 and v.2.0 (*Wang, Gao & Lal, 2000*; *Yuan, Pei & Lai, 2011*), Gromacs v.4.5.5 (*Pronk et al., 2013*) and pharmACOphore (*Korb et al., 2010*) in a seamless rational pipeline developed in Perl/Tcl-Tk. All previously mentioned suites remain a set of numerous modules, lacking

an object-oriented graphical user interface (GUI) to facilitate their use. DrugOn was developed to smoothen and automate the tedious tasks of pharmacophore modeling and 3D structure optimization. In order to provide the user with a 3D molecular viewer, whose usage is a focal point in modern drug discovery and in computer-aided drug design, we also incorporated the Pymol suite (*DeLano, 2002*). The DrugOn idea is to provide a scientifically sound pharmacophore design suite which remains easy to work with and to comprehend. As a result, DrugOn's audience includes both the inexperienced, novice student to the highly demanding researcher and expert in the field of computer-aided drug design. So, by developing a basic interface for novice users we provide an automated platform that will enable them to learn by making easy experiments and to practice in computer-aided drug design by utilizing their ideas and overcoming their lack of experience. On the other hand, DrugOn Pro has a fully integrated interface with all the parameterization an expert needs. More specifically, DrugOn addresses all common problems associated with PDB file formatting and partial charges. Subsequently, the receptor is structurally optimized by energy minimization using a variety of different force fields as implemented into Gromacs. After structural optimization, the Ligbuilder suite is used to generate novel molecules for the given site or to improve an existing family lead or set of compounds. Finally, the pharmACOphore program is used for the pairing of ligands, resulting in the construction of a 3D pharmacophore model.

## PIPELINE'S METHODS AND DESCRIPTION

With a universal installation procedure the DrugOn suite provides the user with two interfaces from which to choose. DrugOn Pro is intended for more experienced users, while the basic, abstract version of DrugOn is intended for inexperienced novice users. A comprehensive flowchart of the DrugOn pipeline can be found in Fig. 1.

### DrugOn

In the main window of DrugOn, the tab layout changes into a frame layout at the left of the main window with two buttons "next" and "previous" (Fig. 2) in order to make the step-by-step process more efficient and the layout smoother for the novice users. It also provides the user with a process log window, at the right of the main window for the real time calculations that take place in the background, with one vertical and one horizontal scroll bars, thus making the information that the user provided easier to traced back. In the DrugOn pipeline, the process for a pharmacophore modeling experiment is broken down to four steps.

(1) *Input preparation*

This is the first and very essential step, often missing from many major suites, where the input (PDB) files are automatically checked and repaired so that all compatibility issues are addressed and basic chemical information is calculated before the experiment. In addition, the missing hydrogens are added and partial charges are calculated. However, in order to make the process easier for the novice users, the choices to remove heteroatoms, for the force field, and to neutralize or not the C' and

Peer J

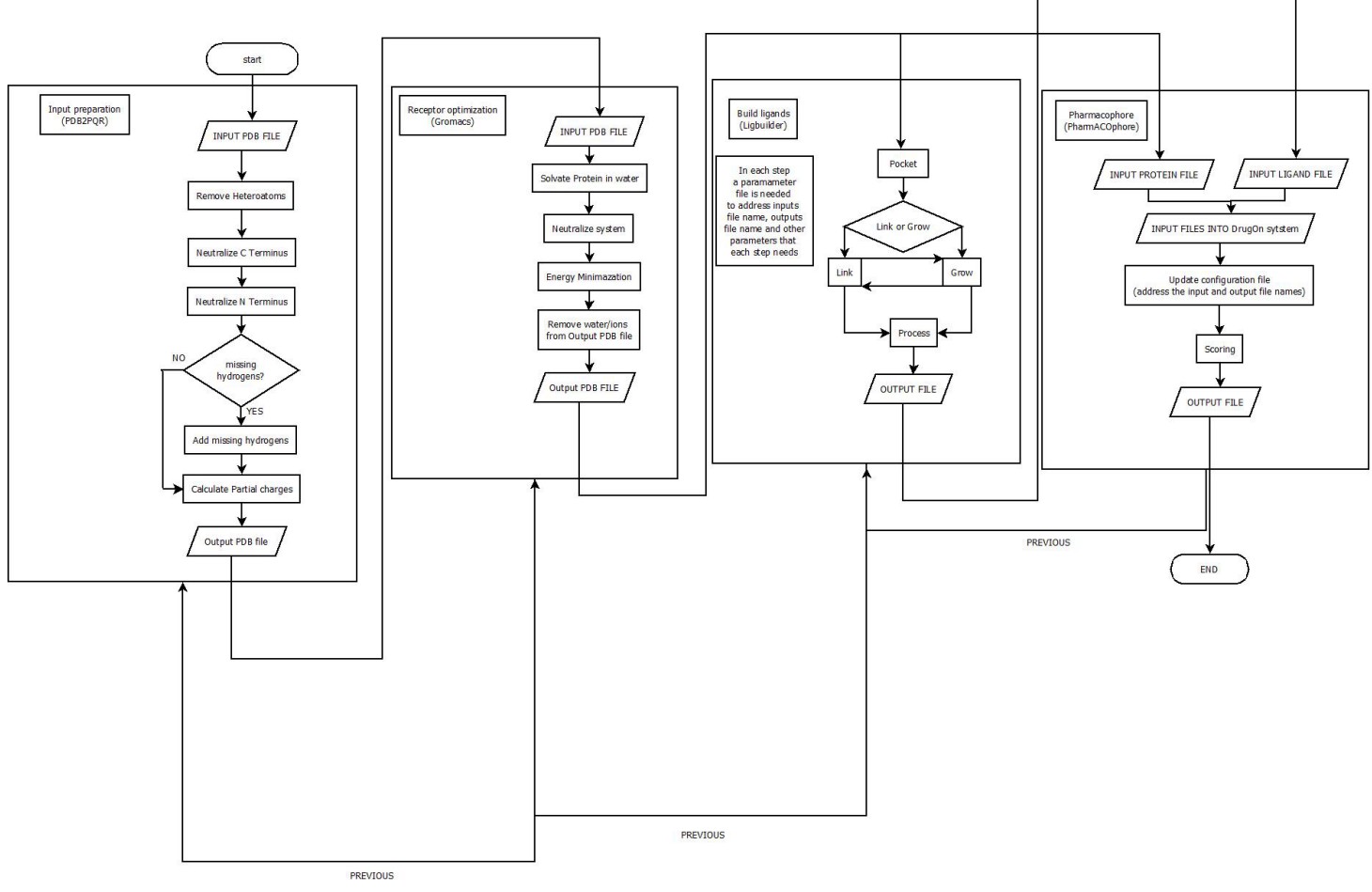

**Figure 1 A flowchart of the DrugOn pipeline.**

N' termini of the protein have been selected by default. Therefore, the responsibility of the user is only to choose the input PDB file, as well as the name and the path of the output PDB file. The above options are processed with the PDB2PQR (*Dolinsky et al., 2007*; *Dolinsky et al., 2004*) algorithm.

(2) *Receptor optimization*

A major problem when removing heteratoms or ligands (Input preparation) from PDB files is that the receptor structure remains in its bound conformation unless it is structurally optimized. In this second automated step, the user can benefit from the conformational optimization of the receptor, an issue that is a major drawback of many structure-based drug designing algorithms. Many inconsistencies and free energy issues that may result from the removal of heteratoms, without bringing it back to the relaxed conformation of the PDB receptor file are addressed. So by using the versatile Gromacs (*Pronk et al., 2013*) suite, the receptor is conformationally optimized via energy minimization before the experiment. Also in this step, the available choices for the user are the input PDB file and the name and the path of the output PDB file.

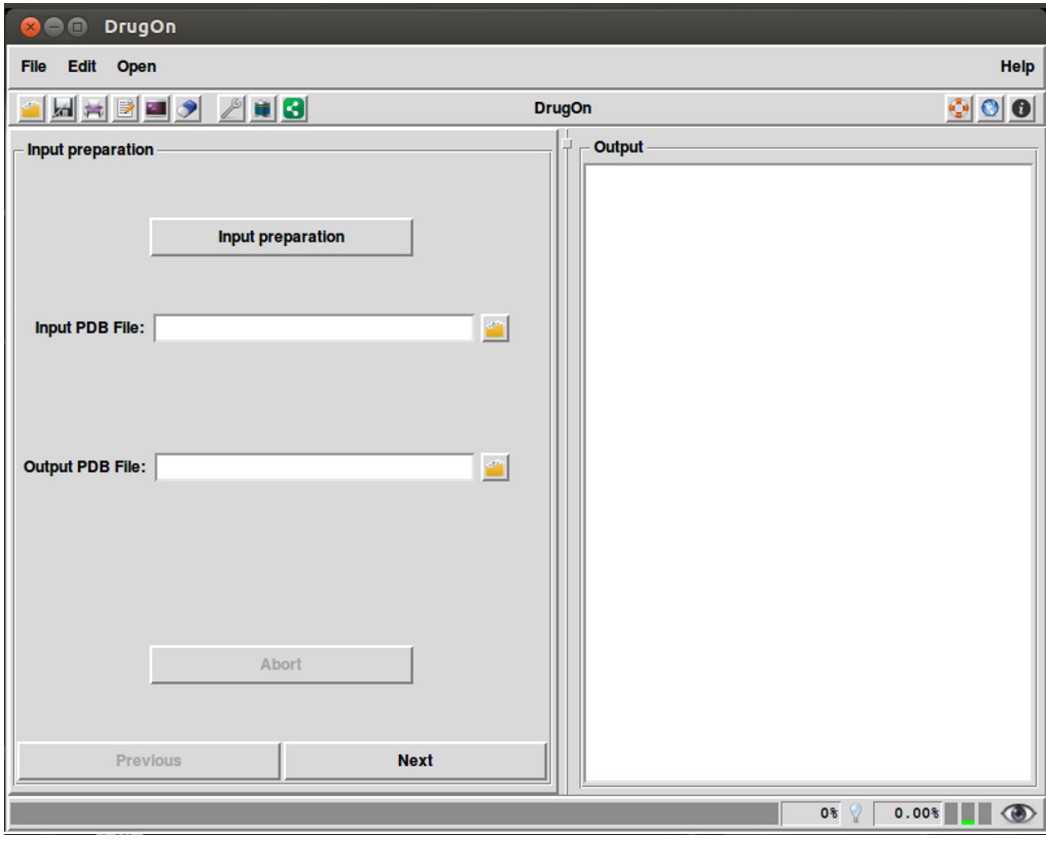

**Figure 2** The main window of the DrugOn platform.

(3) *Ligand building*

At this stage, the actual structure-based drug design of the new ligand structures takes place. This step enables the user to fully parameterize the ligand-building process with the use of Ligbuilder v.1.2 (*Wang, Gao & Lal, 2000*). The user can define the active or pocket of interest by positioning a 'seed' chemical structure in it. The algorithm will then proceed with either the growing of the seed to a drug-like compound, using the predefined criteria in the parameters window or the linking approach (for multiple seeds). Finally, the drug candidates that do not comply with the user's criteria will be screened out by applying a similarity cutoff filter that is user configurable.

(4) *Pharmacophore*

The final step of the DrugOn pipeline is the automatic structure alignment of the molecules that were produced in the previous steps. At this point, the similarity-based scoring function tuned for ligand-based pose prediction is combined with a hybrid ant colony optimization algorithm via pharmACOphore (*Korb et al., 2010*). The scoring function combines an intraligand potential with the distance-dependent potential. The description of molecular similarity is based on hydrogen bond donors and acceptors as well as ring systems and other pharmacophoric features. The identification of corresponding pharmacophoric features in this method depends on the accuracy of the scoring function.

Therefore, a fully parameterized configuration file has been created in order to serve the pharmacophore modeling experiments.

## Toolkits description

A manual for the use of DrugOn is provided in DrugOn's interface using the system's default web browser through the help button. Options such as 'print,' 'clear,' 'save' and 'load all output files' are provided so that the user can print or save for further analysis, trace errors and load previous experiments. Finally, there is an option to clear the output of the log process window in order to start a completely new experiment without any trace of previous outputs that have no longer any use and might be confusing and time-consuming for the user to manually edit. All those options that were introduced earlier have keyboard shortcuts that can be found at the File button on the window's top left corner for faster and more ergonomic use. Additionally, a button for opening a new terminal is available in case the user needs two or more terminals for other uses (besides DrugOn) when an experiment takes place, as the terminal from which the DrugOn was launched is occupied until the user exits DrugOn. The 'handle databases' button allows the user to view and edit molecule databases from the window that pops up. The format that is supported is based on the one used by Ligbuilder to manage fragment and molecular databases. Every database is a folder consisting of the included molecule files in .mol2 format and an INDEX text file which lists the molecular parameters alongside extra information and properties bound to each molecular entry. There is also a 'preferences' button through which the user can manage some of the DrugOn settings, like module path settings and the system's local folder management. The software paths and installation sites are user defined at the DrugOn automated setup by default. Another frame in the preferences window contains the default parameter files, where the user can set the default parameter file that will be used for any given experiments. These files are stored/saved and can be re-used as recipe files to re-run similar experiments by altering the input files. Moreover, an experiment preparation log is saved in the form of a lab-book (with the experimental parameters of importance to pharmacophore design) next to a recording of the input files, the date, and the name of the computer which was used to run the simulation. This way, troubleshooting becomes easy when things go wrong and the chances of finding what went wrong increase dramatically. In the preferences window the user has the option to choose the preferred applications for the text editor, terminal molecular viewer and xvg graph viewer. Additionally, DrugOn's pipeline is capable of starting a log file from the preferences menu that the user predefines. DrugOn will automatically save all output results from the experiments that take place in the form of plain text file format for future reference. This option is essential for keeping track of all useful information that is complicated and may take a lot of time for some users. These files are pre-formatted and ready to print, email or convert into PDF format. Notably, the status tray area provides the user with 2 progress indicators, a progress bar and a percentage (%) of the completed work. A processor memory and swap file usage gauge is to be found right next to the logging indicator, providing real time information of systems resources.

**Peer**J

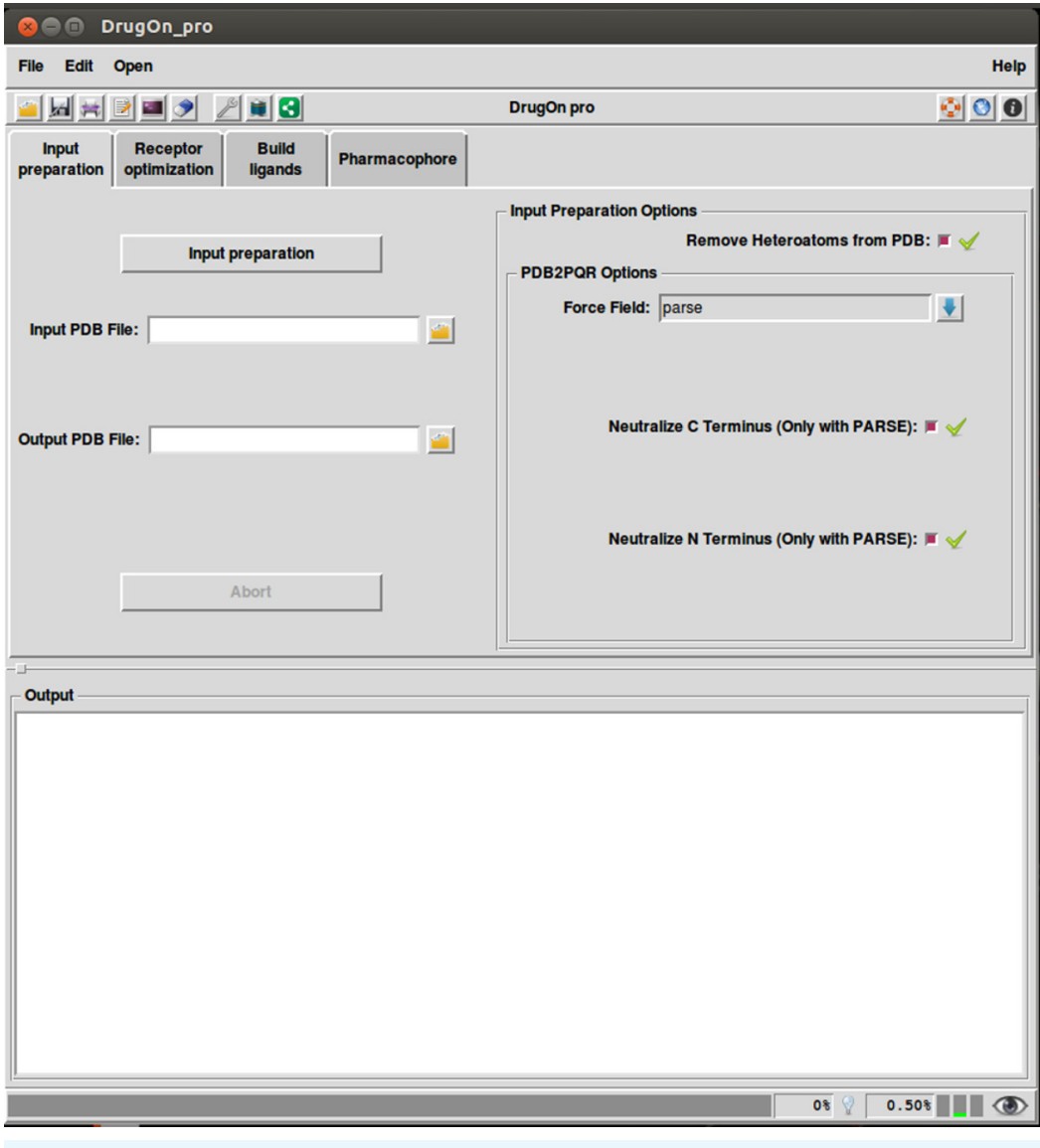

**Figure 3** The main window of the DrugOn Pro platform.

## DrugOn Pro

DrugOn Pro is a more comprehensive, in-depth approach aimed at expert and professional users. DrugOn's Pro main window is a menu interface with a tab step-by-step layout (Fig. 3). It also provides the user with a process log window at the bottom of the main window for the real time calculations that take place in the background, with one vertical and one horizontal scroll in order to make the information that the user provided easily traced back.

In the DrugOn Pro interface for a pharmacophore modeling experiment, the process is separated into the four same steps as DrugOn, with some differences most likely in parameterization.

(1) *Input preparation*

This where the user has the can fully parameterize the pdb input file with choices such us: removing heteroatoms, choosing the force field, and choosing whether to neutralize or not the C' and N' termini of the protein. The above options are processed with the PDB2PQR (*Dolinsky et al., 2007*; *Dolinsky et al., 2004*) algorithm.

(2) *Receptor optimization*

The second step of DrugOn Pro remains the same as in DrugOn, except that the user has the choice of either using the default parameters or fully customizing the parameters for the experiment. The parameters in this step are: the force field that Gromacs uses (Force Field), the type of periodic box that surrounds the protein (Box Type), the distance parameter that decides the size of the box where dynamics will take place (Sol-Box Distance), the choice to perform energy minimization in the presence or absence of water (Solvate Protein in Water), the water model that is used for water molecules (Water Model), the option to remove the overall charge from the system (Neutralize system), the option to remove or leave the water or ions in the output PDB File (Remove water/ions from output PDB File), the option to show a graph of the protein's potential energy MDRun (Show resulting Energy Graph) and to path the parameter file needed for energy minimization (Parameter File) available for the user.

(3) *Build ligands*

At this stage, the actual structure-based drug design of the new ligand structures takes place. DrugOn Pro enables the user to fully parameterize the ligand-building process, with the use of not only Ligbuilder v.1.2 but also Ligbuilder v.2.0 (*Wang, Gao & Lal, 2000*; *Yuan, Pei & Lai, 2011*).

• Ligbuilder v.1.2: The use of Ligbuilder v.1.2 is the same as in DrugOn; the user still has the options of pocket, grow, link and process, but also has the option of Ligbuilder v.2.0.

• Ligbuilder v.2.0: When using Ligbuilder 2.0 the cavity is automatically detected. In the case of many potential active sites, the user will be asked to choose one. The parameters set in the Parameter and Index files are used to start the drug design process. At this step the process is organized into three fully user-customizable phases. First, it prepares and summarizes the 3D properties of the scaffolding, common core structures that later will be generated and analyzed. Then the user has to choose between the growing and linking algorithms of Ligbuilder as soon as the parameters setup section is completed, and then the combination of molecular fragments starts automatically. Finally, the elite molecules are selected for the next step in the compound screening function.

(4) *Pharmacophore*

Identification of corresponding pharmacophoric features in pharmACOphores method counts on the accuracy of the scoring function, and at the final step the modeling experiments of the DrugOn Pro user are provided with two more options. So, the user has the choice of a fully parameterized configuration file that pharmACOphore uses (the default that DrugOn). Moreover, the user is provided with

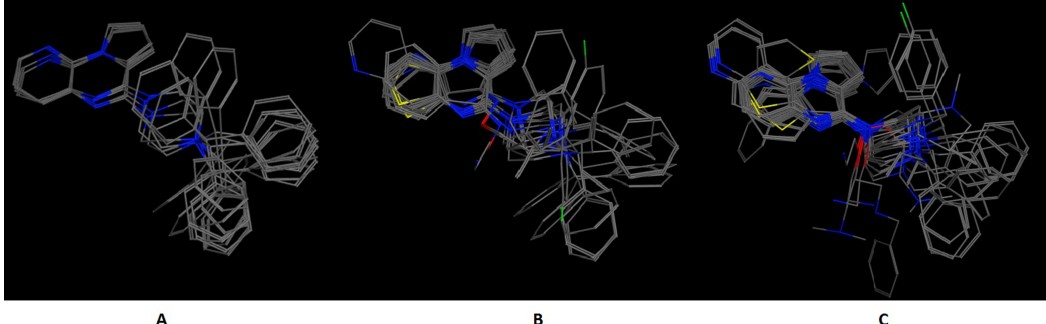

**Figure 4  The 5-HT1B-BRIL use case benchmark of DrugOn.** Here is the 3D alignment of the qualifying molecules for the given receptor. (A) The MOE result, (B) The Schrödinger result and (C) the DrugOn result.

the option to create/edit his own configuration file with the parameters that are needed for each experiment.

A major issue with most major drug design/pharmacophore suites is the installation process on UNIX/Linux based systems, as the command line is not very popular with the majority of users. This is especially true for people who only use graphically enabled operating systems and avoid using applications or software package that runs on Linux because of its difficulty when the graphical interface is not an option. The DrugOn is a pipelined software package based on Linux-Ubuntu systems and has been specifically designed to provide the user with a seamless setup via a graphical interface that simplifies the installation.

## VALIDATION

DrugOn is not the first platform designed for pharmacophore modeling. A similar pipeline approach for a complete drug design toolkit (not pharmacophore) has been published by *Vlachakis et al. (2013a)* with the Drugster toolkit. Moreover, a series of different approaches have been made in the past few years which resulted in commercially available suits like *Moe (2010)*, or some free available suits like pharmer (*Koes & Camacho, 2011*) and open3dqsar (*Tosco & Balle, 2011*); two efficient software packages. Also, Schrödinger has developed PHASE which is distributed as a commercial module of the Maestro suite (*Dixon et al., 2006*; *Dixon, Smondyrev & Rao, 2006*).

In an effort to quantitatively and qualitatively evaluate the performance of DrugOn, we used two different and quite diverge use cases. The first use case is the crystal structure of the chimeric protein of 5-HT1B-BRIL, pdb entry: 4IAR (Fig. 4) and the second case is the pharmacophore design for PARN (Fig. 5) (*Vlachakis et al., 2012*). As a benchmark control we compared DrugOn to the rather expensive and commercially available package MOE and its build-in modules (BREED) and then to the Schrödinger suite and its built-in pharmacophore module PHASE. The results have been summarized in Figs. 4 and 5. It is clear that in both cases the DrugOn suite performed as well as the more expensive rival commercial suites. The number, structure and 3D alignment of candidate compounds and 3D pharmacophore model design as it was produced by DrugOn is almost identical

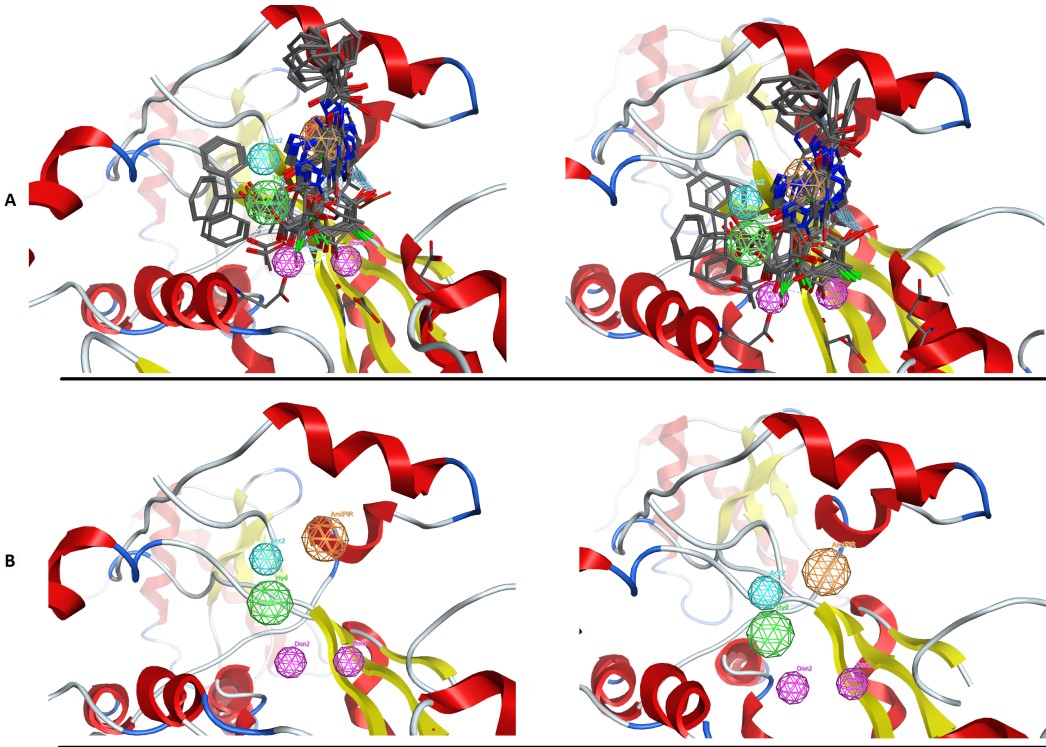

**Figure 5 The PARN use case benchmark of DrugOn.** (A) The 3D alignment of the qualifying molecules for the catalytic site of PARN. On the left is the MOE output while on the right is the DrugOn result. (B) The final 3D pharmacophore model for PARN. The MOE output is on the left while the DrugOn 3D pharmacophore is on the right. The results are almost identical and have been confirmed *in vitro* by enzymatic biological assays.

to that of MOE and similar to PHASE. As far as accuracy and reliability goes, we are now confident that DrugOn reported a set of pharamcophore models that has been evaluated and confirmed by *in vitro* assays, as the predicted poly-A-DNP was found active in the sub-milimolar range (*Vlachakis et al., 2012*).

## CONCLUSION

DrugOn has been developed with the aim to pipeline some of the major drug design suites in an effort to create reliable 3d pharmacophore models. It stands out from its competition by being able to seamlessly combine the results of state-of-the-art algorithms and suites which are just difficult to combine and install or run individually, while remaining distributed as freeware. Operation manuals, tutorials on various use cases, quick guides for teaching purposes as well as multimedia/video installation guidelines and scientific support for DrugOn are provided via our dedicated webserver at http://www.bioacademy. gr/bioinformatics/drugon/.

### Funding

This work was partially supported by (1) The BIOEXPLORE research project (BIOEXPLORE research project falls under the Operational Program "Education and Lifelong Learning" and it is co-financed by the European Social Fund (ESF) and National Resources), and by (2) the European Union (European Social Fund—ESF) and Greek national funds through the Operational Program "Education and Lifelong Learning" of the National Strategic Reference Framework (NSRF)—Research Funding Program: Thales. The funders had no role in study design, data collection and analysis, decision to publish, or preparation of the manuscript.

### Grant Disclosures

The following grant information was disclosed by the authors:
The BIOEXPLORE research project.
European Union.
Greek National Funds.

### Competing Interests

The authors declare there are no competing interests.

### Author Contributions

- Dimitrios Vlachakis conceived and designed the experiments, performed the experiments, analyzed the data, wrote the paper, prepared figures and/or tables, reviewed drafts of the paper.
- Paraskevas Fakourelis performed the experiments, analyzed the data, wrote the paper, prepared figures and/or tables.
- Vasileios Megalooikonomou and Sophia Kossida conceived and designed the experiments, wrote the paper, reviewed drafts of the paper.
- Christos Makris wrote the paper, reviewed drafts of the paper.

### Supplemental Information

Supplemental information for this article can be found online at http://dx.doi.org/10.7717/peerj.725#supplemental-information.

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
