# Peer review of "DrugOn: a fully integrated pharmacophore modeling and structure optimization toolkit"

_PeerJ, doi:10.7717/peerj.725_

## Round 0.1 · original submission · Major Revisions

· Academic Editor

Major Revisions

Thanks a lot for considering PeerJ for the publication of your research. Due to a lack of reviewers and considering my own expertise in the field of Computer Based Drug Design (CBDD), I decided to act as the second reviewer for your work. Despite the work being interesting, and in agreement with the other reviewer, I consider that major revisions needs to be performed on the current manuscript in order to be published in PeerJ.

Above all, my major concerns are regarding the validity of the application of DrugOn methods. In spite that DrugOn is not proposing a method itself but orchestrating different methods, produced by other authors, the validity of its application needs to be properly addressed. I suggest to the authors to use some databases like ZINC, Diversity Dataset and/or the Ligand PDB in order to define, using congeneric dataset, gold standards and decoys, including true and false positive molecules. Due to the huge number of molecules available within those databases, that task should be easily performed.

In addition, some corrections to the text should be also implemented. As an example, there some occasions in which the authors compare DrugOn with other methods using qualitative expressions that are not numerically supported. In the field of CBDD we have the unique opportunity to use qualitative approaches (like a pharmacophore definition) but relying in quantitative metrics. For instance, when comparing the performance of different methods, the authors should use ROC curves or at least confusion matrices. In the current version of this work there are none quantitative measures that can be used for comparisons. Despite these issues and agreeing with the reviewer, there is a lack of freely available methods/tools to produce pharmacophore models and therefore the work presented in this paper could be of broad interest to the CBDD community. However, the free availability of a set of tools is not necessarily a scientific contribution but a technical one. The authors should make a compelling effort to demonstrate that DrugOn is more than just a technical achievement. To do so, a proper evaluation, using a quantitative approach, must be included to be considered for publication.

Last but not least, a thoroughly revision by an English native speaker should be performed to improve clarity and soundness.

Reviewer 1 ·

Basic reporting

The Manuscript form Vlachakis et al, reports on the development of a novel streamline application for the generation of pharmacophores. The basic concept of the application is interesting, but I believe it will require a major rewriting before it could be considered fro publication. In particular:

The introduction is very confusing. It is not clear in what context the application was developed and what was the original rationale behind it. The very first paragraph (line 26 to 34) is ambiguous. There is no evidence to support that "current methods have almost exhausted the range of their possibilities". In fact, it is possible to prove the opposite by looking at the current literature. I would suggest to remove this paragraph altogether. The rest of the introduction should only cover, in a concise manner, the background on the pharmacophore concepts, the current pharmacophore applications and provide a rationale for the design of DrugOn. The discussion and conclusion should follow the same, logical, flow. Also, a figure representing the DrugOn procedure, as flowchart should be included.

Experimental design

The concept presented is interesting. However, a validation of the proposed application should be provided. I would recommend a set of basic simulations were DrugOn is compared with other, already available, software packages (the statement "However, DrugOn is not inferior to these suits that we listed before." needs to be supported with quality data).

Validity of the findings

As stated before, DrugOn could be a useful application, but at this point it is not clear how this would perform against other packages. Also, it is not clear what is the final output of DrugOn is. It is clear that it would be a pharmacophore model, but a pharmacophore is often the starting point of a drug design exercise. How usable is the DrugOn output? would DrugON generate a single pharmacophore model? All these questions need to be addressed.

---

## Round 0.2 · Minor Revisions

· Academic Editor

Minor Revisions

Thanks a lot for submitting an improved version of you manuscript addressing the issues raised in the previous revision. In its current form, the manuscript should be interesting for PeerJ readers and therefore is almost ready for publication. However, one minor issue raised by the reviewer (see below) should be addressed before publication. On the other hand, I'd like to encourage the authors to consider supplementary figures S1 to S3 as part of the main document. From my point of view, the manuscript will be importantly improved by including these figures in the main text. In doing so, workflow and validation will be properly highlighted. While the former is the most important knowledge repository (workflow) to be transferred to the users, the later is the demonstration that the software is producing results comparable to those of the state of the art.

Reviewer 1 ·

Basic reporting

The manuscript has been substantially improved from the first submission and it should be considered for publication after a very minor modification. To improve clarity of figure S3, the authors should use the same style (either showing the surface or not; no hydrogen displayed) for all the 4 images.

Experimental design

-

Validity of the findings

-

Comments for the author

-

---

## Round 0.3 · accepted · Accept

· Academic Editor

Accept

Thanks a lot for addressing all the reviewers' comments. The manuscript is now ready for publication. We hope to receive your future contributions and... merry Xmas!